# Advanced Biomaterials for Lacrimal Tissue Engineering: A Review

**DOI:** 10.3390/ma17225425

**Published:** 2024-11-06

**Authors:** Kevin Y. Wu, Archan Dave, Patrick Daigle, Simon D. Tran

**Affiliations:** 1Department of Surgery, Division of Ophthalmology, University of Sherbrooke, Sherbrooke, QC J1G 2E8, Canada; yang.wu@usherbrooke.ca (K.Y.W.);; 2Department of Medicine, University of British Columbia, Vancouver, BC V6T 1Z3, Canada; 3Faculty of Dental Medicine and Oral Health Sciences, McGill University, Montreal, QC H3A 1G1, Canada

**Keywords:** biomaterials, tissue engineering, lacrimal gland, dry eye disease, natural and synthetic biomaterials, 3D bioprinting, regenerative medicine

## Abstract

The lacrimal gland (LG) is vital for ocular health, producing tears that lubricate and protect the eye. Dysfunction of the LG leads to aqueous-deficient dry eye disease (DED), significantly impacting quality of life. Current treatments mainly address symptoms rather than the underlying LG dysfunction, highlighting the need for regenerative therapies. Tissue engineering offers a promising solution, with biomaterials playing crucial roles in scaffolding and supporting cell growth for LG regeneration. This review focuses on recent advances in biomaterials used for tissue engineering of the lacrimal gland. We discuss both natural and synthetic biomaterials that mimic the extracellular matrix and provide structural support for cell proliferation and differentiation. Natural biomaterials, such as Matrigel, decellularized extracellular matrices, chitosan, silk fibroin hydrogels, and human amniotic membrane are evaluated for their biocompatibility and ability to support lacrimal gland cells. Synthetic biomaterials, like polyethersulfone, polyesters, and biodegradable polymers (PLLA and PLGA), are assessed for their mechanical properties and potential to create scaffolds that replicate the complex architecture of the LG. We also explore the integration of growth factors and stem cells with these biomaterials to enhance tissue regeneration. Challenges such as achieving proper vascularization, innervation, and long-term functionality of engineered tissues are discussed. Advances in 3D bioprinting and scaffold fabrication techniques are highlighted as promising avenues to overcome current limitations.

## 1. Introduction

The lacrimal gland is essential for ocular health, as it is responsible for producing the aqueous component of the tear film that lubricates the eye, providing corneal nutrients, and protecting against microbial infections. Dry eye disease (DED) affects approximately 344 million people globally, with prevalence estimates varying widely because of differences in study populations and definitions. General DED prevalence is estimated at 5–50%, with higher rates in populations over 50 years of age and in women due to hormonal factors [1]. In the USA, it is estimated that about 20 million people suffer from DED, with a higher incidence in urban settings due to environmental factors like air pollution and lower humidity [1]. Dysfunction of the lacrimal gland leads to aqueous-deficient DED, a condition characterized by symptoms such as ocular discomfort, visual disturbances, and a significant reduction in quality of life [2]. Current treatments for DED primarily focus on symptom management—using artificial tears, anti-inflammatory agents, and punctal plugs—without addressing the underlying glandular dysfunction. These approaches often offer only temporary relief and may have side effects that affect patient compliance [3,4].

Advancements in tissue engineering present a promising avenue for restoration of lacrimal gland function by regenerating damaged tissue and reinstating natural tear production. Central to this approach is the development of suitable biomaterials that can serve as scaffolds to support cell growth, differentiation, and the formation of functional glandular structures [5]. Both natural and synthetic biomaterials have been explored for this purpose. Natural biomaterials, like Matrigel, decellularized extracellular matrices, chitosan, silk fibroin hydrogels, and human amniotic membrane, have shown potential in mimicking the extracellular matrix and promoting cell viability and differentiation. Synthetic biomaterials such as polyethersulfone, polyesters, and biodegradable polymers like poly-L-lactic acid (PLLA) and poly(lactic-co-glycolic acid) (PLGA) offer tunable mechanical properties and structural versatility but may require modifications to enhance biocompatibility and cellular interactions [6].

This review focuses on recent advances in biomaterials used for lacrimal gland tissue engineering. We examine the properties of various natural and synthetic biomaterials, their applications in supporting lacrimal gland cell growth, and their roles in facilitating tissue regeneration. Additionally, we discuss the integration of stem cells and growth factors with these biomaterials to enhance regenerative outcomes. Challenges such as replicating the complex architecture of the lacrimal gland, achieving proper vascularization and innervation, and ensuring long-term functionality of engineered tissues are also addressed. By providing an overview of the current state of biomaterial-based lacrimal gland engineering, we aim to highlight the potential of these approaches to develop effective regenerative therapies for patients with lacrimal gland dysfunction.

## 2. Lacrimal Gland Anatomy and Disorders

The aqueous portion of the tear film is secreted by the lacrimal gland (LG), which sits on the anterior, superotemporal portion of the orbit within the lacrimal fossa of the frontal bone [7]. The LG comprises both the main lacrimal gland and smaller accessory lacrimal glands of Krause and Wolfring (Figure 1). The glands of Krause are located in the conjunctival fornices, and the glands of Wolfring are distributed along the nonmarginal tarsal borders. Tear fluid released by the glands help to nourish, moisten, and protect the ocular surface through its inherent antibacterial activity. Aqueous tears consist mainly of water, electrolytes, and proteins responsible for maintaining the health of the ocular surface [8]. This 2 cm long tubuloacinar gland contains many lobules, which are composed of mixed serous and mucinous acini, as well as supporting myoepithelial cells and between 8 and 12 main excretory ducts [7]. The polarized lacrimal epithelial cells are organized into distinct acini, which encircle a central duct lined by ductal epithelial cells. Acinar cells compose approximately 80% of the cell population, while duct cells account for approximately 10 to 12% of the total [9]. The LG is anchored to the orbital periosteum by connective tissue, receives blood from the lacrimal artery, and is innervated by the lacrimal nerve through the ophthalmic branch of the trigeminal nerve (CN V). The LG also receives sympathetic innervation, which works in conjunction with parasympathetic signals to regulate fluid secretion and maintain homeostatic tear volumes [9]. Myoepithelial cells around the acini contract in response to nerve signals, allowing for fluid to drain onto the ocular surface through the excretory duct. Neural regulation of the LG is a complex process, mainly governed by the tear–reflex loop pathway which secretes tears in response to ocular stimulation [9].

The lacrimal gland plays a crucial role in ocular health by producing the aqueous component of the tear film, which is essential for lubricating the eye, providing nutrients to the cornea, and protecting against microbial infections. Loss of lacrimal gland function leads to a deficiency in tear production, resulting in dry eye disease (DED). DED is a multifactorial condition characterized by a variety of non-specific symptoms, including ocular redness, burning sensations, itching (i.e., pruritus), foreign body sensation, and the presence of stringy or mucous discharge [2]. If left untreated, DED can progress to severe complications, such as corneal epithelial breakdown, ulceration, and corneal melting, which significantly threaten visual acuity [10]. Additionally, the chronic discomfort and visual disturbances associated with DED can markedly diminish an individual’s quality of vision and overall quality of life [11]. There is a high prevalence of DED, at around 12%, depending on sex, age, and location [12]. In short, alterations in the secretory function of the lacrimal gland can compromise tear film stability and degrade the quality of tear secretions, leading to a disruption in ocular surface homeostasis. Inflammation (i.e., dacryoadenitis) secondary to autoimmune diseases, such as Sjögren’s syndrome [13], rheumatoid arthritis, and systemic lupus erythematosus, can be associated with LG dysfunction. Sjögren’s syndrome is a rare, T-cell-driven disease characterized by lymphocytic infiltration of the lacrimal and salivary glands. It should be noted that there is no standardized screening tool to refer to for Sjögren’s syndrome; therefore, it is often underdiagnosed [14]. Beyond autoimmune conditions, factors such as chemotherapy, radiation therapy, and graft-versus-host disease can damage lacrimal gland tissue or provoke immune-mediated attacks on the gland. Physical trauma—whether surgical or accidental—can disrupt tear secretion pathways. Other causes of LG dysfunction include but are not limited to infectious agents (bacterial, viral, and fungal), neoplastic processes (benign or malignant tumors), granulomatous diseases like sarcoidosis, infiltrative diseases such as amyloidosis, and age-related atrophy of the gland [11,15].

Current treatments for dry eye disease, including those caused by LG dysfunction, primarily focus on symptom management rather than addressing the underlying lacrimal gland dysfunction. Conservative approaches include moisture-chamber spectacles to reduce tear evaporation and the frequent application of artificial tears to supplement tear volume [16,17]. More advanced therapies involve punctal plugs to decrease tear drainage, topical anti-inflammatory agents, such as cyclosporine A and lifitegrast, to modulate ocular surface inflammation, and short-term use of topical corticosteroids during acute exacerbations [18]. Autologous serum eye drops are also employed in severe cases to provide essential tear components. However, these treatments often offer only temporary relief, require frequent administration, and may have side effects that impact patient compliance [3,4]. The limitations of current therapies highlight the need for innovative solutions that restore natural tear production in patients suffering from LG dysfunction. This has sparked growing interest in lacrimal gland tissue engineering as a potential strategy to restore gland function and provide long-term relief for patients with aqueous-deficient dry eye disease.

## 3. Biomaterials for Lacrimal Tissue Engineering

Most commonly, tissue engineering consists of manufacturing a biocompatible scaffold and using specific progenitor cells and multiple growth factors to ensure effective regeneration and reconstruction of tissue. Scaffolds serve as the structural foundation for cultured cells, enabling them to attach and form the unique morphology of the tissue. Moreover, scaffolds must provide a suitable environment for differentiated cells to expand, proliferate, migrate, and be resilient enough to endure foreign stressors. The most ideal scaffold is one that mimics the extracellular matrix (ECM) of the target tissue [6].

However, scaffold-free tissue engineering relies on the self-organization of cells, which are allowed to produce their own ECM through cell-to-cell interactions. This closely mimics natural tissue development, reduces the risk of scaffold-related complications, and offers natural communication among cells in the tissue. Schrader et al., (2009) demonstrated the formation of 3D LG spheroids from acinar cells using a rotary-cell-culture system [19]. However, apoptosis was observed in the spheroid center, likely due to the lack of vascular supply, which consequently diminished cell secretory function by day 28. Ackermann et al., (2015) isolated murine LG stem cells and explored their potential for differentiating into various cell types [20]. The group specifically employed the hanging drop culture method to form 3D organoid bodies; however, the transmission electron microscopy (TEM) results showed that most cells remained undifferentiated with a few exceptions. Lin et al., (2017) demonstrated LG progenitor cell proliferation into a gland-like spheroid and acinotubular structure when cultured in a 3D environment using laminin gel [21]. The cells showed signs of acinar and ductal differentiation, expressing markers such as K4, lactoferrin, and aquaporin 5. However, the size, shape, and exact composition were difficult to control, and the 3D cultures lost their structural integrity over time. While the 3D culture in this study lasted for about 2 weeks, Tiwari et al., (2018) produced lacrispheres lasting for 21–28 days, during which they retained their stemness, proliferative potential, and secretory function. The group enzymatically digested human LG and cultivated them in a serum-free, 3D culture system. The group provided a scaffold-free platform that was able to maintain the long-term survival of LG cells and secrete significant levels of tear protein, including lysozyme, lactoferrin, and sclgA [22]. Despite these advancements, scaffold-free tissue engineering faces challenges, as noted by its inability to maintain long-term structural stability and proliferative capacity as compared to scaffold-based 3D culture systems. However, the ability of these systems to self-organize into functional secretory units demonstrates the regenerative potential of the LG and encourages future exploration into scaffold-free LG restoration.

Scaffold-based systems use either natural or synthetic biomaterials to support LG cell growth and differentiation. These materials are often biocompatible and possess specific mechanical properties and degradation rates, making them adaptable for different tissue-engineering applications. The spatial arrangement of cells, delivery of growth factors, size, and shape can also be controlled through scaffold-based systems. The most popular natural biomaterial for organoid development is Matrigel; however, its variability in composition among batches, indeterminate degradation rate and immunogenic properties presents challenges in tissue transplantation applications [5]. Kozlowski et al., (2021) provide extensive alternative solutions, such as decellularized ECMs, synthetic hydrogels, and gel-forming recombinant proteins [5].

This section covers both natural and synthetic biomaterials that are used, currently investigated, or may be considered in LG tissue engineering.

### 3.1. Natural Biomaterials

Xenografting refers to the transplantation of tissue from animals to humans. This approach to bioengineering provides an immediate source for grafting, hence significantly reducing waiting times. It typically offers fully formed tissues with proper structure and functional vascular components in place. For example, Henker et al., (2013) claim that the porcine LG anatomy shares morphological similarities with human glands, albeit larger in size and having more seromucous secretory units compared to humans [23]. In fact, the team proposes that pig-to-human LG transplantation could be feasible, owing to the straightforward vascular connections and favorable anatomical positioning. However, the study describes immunological barriers and the risk of endogenous pig retrovirus transfer as obstacles to pig-to-human transplantation. While rat, mouse, and rabbit models are often compared to human LGs, they each possess unique characteristics that make interspecies transplantation challenging. However, exploration of LG models from other species are critical in understanding LG function, pathologies, and developing bioengineering tools [24].

Many natural biomaterials are considered for LG bioengineering [Table 1]. Matrigel is a natural polymer derived from ECM proteins secreted by Engelbreth–Holm–Swarm (EHS) mouse sarcoma cells. Matrigel contains many structural proteins, such as laminin, collagen IV, entactin, and heparin sulfate proteoglycans, hence holding major components of the many tissue basement membranes [25]. Many growth factors, such as epidermal growth factor (EGF), fibroblast growth factor (FGF), and insulin-like growth factor 1 (IGF-1), are included in Matrigel [26]. These activate cellular signaling pathways, including mitogen-activated protein kinase (MAPK) and PI3K/AKT pathways, and, ultimately, stimulate mitosis and cell proliferation. In fact, stem cell differentiation is enhanced in vitro when plated on the thick basement membrane matrix due to its ability to promote cell survival, differentiation, and vascularization [27]. Early adopters of Matrigel, such as Schechter et al., (2002), applied its use as a “raft culture” of lacrimal acinar cells enclosed within the gel coating [28]. These rafts were seeded onto Matrigel-coated culture plates, and growth was observed for 28 days. Matrigel closely mimicked the natural environment of the LG and allowed the cells to maintain a differentiated state and polarity, enabling better in vitro study of the acinar physiology. However, the study observed a decrease in the expression of prolactin (PRL) and MHC-II proteins after 21 days in culture, suggesting that while Matrigel supports acinar cell development for a significant period, there may be a gradual loss of functionality over time, limiting clinical translation or long-term uses. Furthermore, Matrigel possess rather poor mechanical properties, compared to other synthetic and natural biomaterials; hence, it is usually fortified with another material, such as collagen, if used as a scaffolding material [29]. However, Matrigel’s laminin and collagen IV components enable integrin binding and is critical for mechanotransduction, which allows cells to sense and response to the matrix and guide differentiation [30]. Tiwari et al., (2012) cultured human LG tissue on different substrates, such as Matrigel, collagen, and human amniotic membrane, and evaluated the expression of stem cell (ABCG2, c-Kit, and ALDH1) and differentiation markers [31]. The functionality of cultured cells was also assessed through secretion of secretory proteins, such as IgA, lactoferrin, and lysozyme. Matrigel provided an optimal substrate and exhibited higher rates of proliferation while maintaining secretory function. However, the study notes that the differences between Matrigel, collagen, and human amniotic membrane were not significant. Interestingly, Yoshino (2000) found lower rates of proliferation on Matrigel when compared to type 1 collagen gel with and without fibroblasts and plastic as the control. However, Matrigel promoted higher levels of acinar differentiation [32,33]. The incorporation of fibroblasts in collagen gels also further supported ductal differentiation. Asal et al., (2023) aimed to develop functional LG organoids using human-induced pluripotent stem cells (iPSCs) through a multizonal ocular differentiation approach [34]. iPSCs were seeded onto a Matrigel matrix to support their differentiation over the course of 7 weeks. The study found that Matrigel effectively supported the formation of acinar structures and facilitated branching morphogenesis. Gleixner et al., (2024) demonstrated the use of Matrigel to develop immortalized human LG cells lines in a 3D spheroid culture model [35]. The group claimed Matrigel’s complex composition of basement membrane components, such as laminin, collagen IV, and other growth factors, are effective in enhancing differentiation and promoting gland-like structure formation. Zeng et al., (2024) utilized Matrigel to develop a culture system to support the in vitro expansion of mouse LG epithelial cells [36]. Matrigel was used in conjunction with two small molecules, Y27632 and SB431542, to promote high proliferation and maintain cell morphology through multiple passages. Upon removal of the small molecules, LG epithelial cells were differentiated into secretory cells with increased expression of differentiation makers such as AQP5 and lactoferrin (LTF). The authors demonstrate that Matrigel effectively mimics the ECM in vitro and, when combined with small molecules, offers a reliable strategy to promote the proliferation of LG epithelial cells. Overall, Matrigel appears to be the most popular scaffold material for LG bioengineering and current trends seem to point toward fortifying Matrigel and integrating growth factors, small molecules, cytokines, and activating transcription factors to enhance cell proliferation and differentiation.

Wiebe-Ben Zakour et al., (2024) recently demonstrated the use of decellularized LG hydrogel derived from porcine decellularized LGs as a bioink to study aqueous deficient DED in vitro. The study showed increased cell viability and proliferation compared to the traditional substrates, such as collagen type 1 and Matrigel. The group suggests that the newly formed hydrogel maintains the native biochemical composition of regular LG ECM; however, rapid degradation poses challenges for long-term cultivation and limits various tissue-engineering applications [37]. The biological mechanism by which decellularized ECM supports LG engineering lies in its retention of bioactive molecules, such as glycosaminoglycans and growth factors, which help guide native cell behaviors essential for LG function. The group further aimed to enhance the mechanical stability and reduce the rapid biodegradation of the hydrogel derived from decellularized porcine LG (dLG-HG). Varying concentrations of genipin was used to crosslink dLG-HG, and the degradation was quantified over 10 days through the activity matrix metalloproteinases. A 0.5 mM concentration of genipin increased the stiffness of dLG-HG by about 46% and significantly delayed the cell-dependent biodegradation of the biomaterial, without compromising cell viability and secretory function. Genipin can be used to control and fine-tune the degradation of the hydrogel to match the requirements of specific tissue-engineering investigations, hence providing sufficient time for tissue modeling and regeneration without the material breaking down too quickly. Lin et al., (2015) prepared an LG scaffold through decellularization of adult rabbit LG. Adult rabbit LG progenitor cells were cultured and seeded on the scaffold resulting in good cell viability and function [38]. Massie et al., (2017) aimed to evaluate decellularized porcine jejunum (SIS-Muc) as a potential scaffold for the reconstruction of LG tissue [39]. SIS-Muc was found to retain the critical basement membrane proteins such as collagen IV and laminin after decellularization. Furthermore, LG epithelial cells cultured on SIS-Muc were found to proliferate more rapidly and remodel the mucosa into a thicker cell layer. In addition to the increased metabolic activity, there was evidence of cell polarization and presence of secretory vesicles; however, acini-like structures were absent. Furthermore, other key tear film proteins, such as lysozymes, lipocalin-1, and lactoferrin, were found in lower levels than naturally produced in tears. Spaniol et al., (2013) also demonstrated decellularization of porcine LG to develop scaffolding and noted an intact connective tissue matrix, with expression of critical basement component proteins such as collagen IV and laminin [40]. Decellularized LG hydrogel is a promising biomaterial for LG reconstruction; however, unpredictable degradation rates remain a significant challenge, and more mechanical fine-tuning is required before clinical translation.

Chitosan is a natural polysaccharide and is derived from chitin. Deacetylation increases the water solubility of the biomaterial, which allows it to be used in bioengineering applications. The polymer is linked through β-(1→4) glycosidic bonds and allows for the formation of hydrogen and ionic interactions, hence creating scaffolds for growth, attachment, and differentiation of LG cells. Chitosan has a positively charged surface, allowing for better facilitation with negatively charged cell membranes, which promotes cell adhesion. In addition, Chitosan’s antimicrobial effects help reduce inflammation and create a favorable environment for tissue regeneration [41]. Most significantly, chitosan is highly biocompatible, nontoxic, biodegradable, and possess low immunogenicity [42]. Hsiao et al., (2017) utilized chitosan biomaterials to facilitate the regeneration of LG tissue by promoting branching morphogenesis [43]. The group demonstrated that chitosan enhances the temporal and spatial expression of hepatocyte growth factor (HGF)-related molecules and increasing branching of LG tissue. Chitosan was also shown to promote increased binding affinity between HGF and c-MET, activating downstream signaling pathways (MAPK and Akt/PKB) that are vital for branching morphogenesis. The group postulates that the structural similarity of chitosan to glycosaminoglycans allows for it to interact effectively with endogenous growth factors, hence removing the need for external supplements or bioincompatible agents. However, the study indicates that other morphogens may also be involved, and the exact identity and roles of these additional factors was not explored. Further morphogenetic effects of chitosan were also found to be dependent on the molecular weight and integrity of its glycosidic linkages, suggesting the effectiveness of chitosan may be influenced by its specific preparation and chemical properties, hence complicating its standardization for clinical use.

Dai et al., (2022) developed an innovative in situ-forming injectable hydrogel as a degradable lacrimal plug for DED [44]. The hydrogel plug was synthesized based on a methacrylate-modified silk fibroin (SFMA) that was photo-crosslinked in situ using visual light. Silk fibroin is a natural protein derived from silk-producing arthropods and possess strong biocompatible, biodegradable, and mechanical properties [45]. Silk fibroin contains Arg-Gly-Asp (RGD) sequences that bind to integrins on the cell surface and may initiate signaling pathways that promote cell attachment and proliferation [46]. Photo-crosslinking allowed for precise spatial and temporal control, ensuring the hydrogel formed at the desired location within the lacrimal passage and that solidification triggered on demand when the material was exposed to light [47]. This may allow clinicians to create a custom-fit, bioengineered solution for each patient, reducing the risk of plug migration and surgical complications. Moreover, mechanical properties can be controlled by adjusting the duration and intensity of light to suit different clinical needs. The SFMA hydrogel demonstrated excellent biocompatibility both in vitro and in vivo, noting no significant inflammatory responses and improved lacrimal fluid retention [44]. In addition, indocyanine green fluorescence tracer nanoparticles (FTNs) were incorporated into the hydrogel to allow for long-term non-invasive tracking when exposed to near-infrared light. This has large clinical implications in patient monitoring and long-term management of dry eye, enabling the ability to make timely adjustments and reducing complications.

Human amniotic membrane is a natural collagen-based biomaterial derived from the innermost layer of the placenta, specifically from the amnion [48]. The membrane is widely used in medical applications, such as wound healing [49] and tissue engineering for eye, skin, vascular system, urethral, cartilage, bone, nerve, heart, and ENT applications [50]. Amniotic membrane contains a plethora of growth factors and nutrients, which enable intracellular signaling cascades that promote wound healing and epithelization. Schrader et al., (2007) evaluated the growth pattern and secretory function of LG acinar cells cultured on amniotic membrane in rabbit models [51]. The cells were analyzed over 28 days, and the secretory function was tested through β-hexosaminidase activity. Secretory response diminished significantly at the end of 28 days, and the cells showed a reduction in secretory granules, and the formation of flattening and spindle-shaped cell morphologies was observed. Interestingly, cells in direct contact with the amniotic membrane retained their acinar morphology. The group suggests that amniotic membrane has the capacity to support cell viability, differentiation, and secretory function, highlighting its use in regenerative medicine, specifically for the reconstruction of LG tissue. However, Singh et al., (2022) suggest that human amniotic membrane may not adequately support the formation of new blood vessels, which could hinder the growth of acinar cells and lead to cell death. Additionally, amniotic membrane can only grow on surfaces, hence restricting its ability to serve as a true 3D scaffold. Hence, further optimization and investigation are required to elucidate the ability of human amniotic membrane to maintain long-term cell functionality and support the growth and differentiation of many cells in a complex, natural structure [52]. Interestingly, Ogawa et al., (2017) demonstrated that the heavy-chain-hyaluronan/pentraxin 3 (HC-HA/PTX3) protein, which was purified from human amniotic fluid, preserved tear secretion and maintained conjunctival goblet cell density in mice with chronic graft-versus-host disease (cGVHD) [53]. Furthermore, it significantly reduced inflammation through suppression of immune cells and mitigated fibrosis in the lacrimal gland. This study underscores the potential of HC-HA/PTX3, suggesting that the human amniotic membrane could be effectively combined with other scaffolds to promote therapeutic effects.

Chen et al., (2014) used xenogeneic (bovine) acellular dermal matrix combined with autologous conjunctival tissue to reconstruct the lacrimal duct in patients with partial or total absence of the lacrimal duct [54]. The method involved creating a lacrimal duct by rolling a conjunctival petal attached to an acellular dermal matrix into a tube and surgically implanting the structure to form a functional tear drainage system. Epiphora symptoms were alleviated in all human patients over approximately 7.2 months. The acellular dermal matrix produced a strong, stable framework, which prevented collapse and was essential for ensuring the duct’s patency and position. In addition, the matrix was biocompatible and capable of degrading at an optimum pace, allowing the surrounding natural tissues to take over the support function through minimal scarring. However, long-term outcomes are not yet known, so it is unclear how the biomaterial will maintain duct patency and avoid complications such as blockage or fibrosis.

Natural hydrogels, such as collagen, fibrin, and gelatin, may also be used in LG reconstruction. Collagen is one of the most abundant proteins in the body and forms the structural basis of many tissues. Hirayma et al., (2013) utilized collagen gel to encapsulate the bioengineered glands during transplantation into mice with LG defects [55]. The bioengineered LG glands developed proper 3D structures, discharged essential tear proteins, and the harderian glands secreted lipids for tear film stability. This study and Nakamura et al., (1996) highlight collagen I as an important biomaterial in facilitating branching and overall morphogenesis during LG development [56]. In fact, Rusch et al., (2021) investigated the role of freshly isolated mesenchymal stem cells (MSCs) in the LG of mouse models and found that 85% of the MSCs produced collagen type I [57]. However, during allogeneic transplantation, collagen production appeared to contribute to pathogenic fibrosis, which may impair LG function and contribute to DED in the context of chronic graft-versus-host disease. Hence, while MSCs have the potential to aid in tissue repair through collagen secretion, the optimization of collagen secretion is critical to avoid fibrotic outcomes. Moreover, Lee et al., (2016) noticed that collagen type II 1a-based peptides possess anti-inflammatory properties and can improve the pathogenesis of ocular surface disorders by increasing tear volume and stabilizing the corneal epithelium [58].

Fibrin, typically sourced from human plasma, is a less explored biomaterial for LG reconstruction. However, fibrin hydrogels have been used to promote salivary gland regeneration, showing effectiveness in improving epithelial tissue organization while facilitating the development of vascularization and nerve formation [59]. Fibrin actively promotes cell adhesion, migration, and differentiation; however, it often produces insufficient mechanical strength and very fast degradation rates which hinder long-term tissue support [60]. A common theme among natural hydrogels is that they are prone to breaking down by proteases such as matrix metalloproteinases (MMPs); hence, they are often altered with crosslinkers that are able to fine-tune degradation rate [61].

**Table 1 materials-17-05425-t001:** Natural biomaterials suitable for lacrimal gland tissue bioengineering.

Biomaterial	Derivation	Features	Disadvantages	References
Matrigel	ECM proteins of EHS mouse sarcoma	Supports cell proliferation, acinar differentiation, mimics natural basement membrane, and promotes gland-like structure formation	Variability between batches and animal-derived, hence limiting clinical applications; decreased expression of proteins after some time, hence limiting long-term use; indeterminate degradation rate; and may be immunogenic	[28,29,31,34,35,36,37]
Decellularized Lacrimal Gland Hydrogel	Porcine decellularized lacrimal gland, most commonly	Maintains the native biochemical composition of the lacrimal gland ECM	Rapid degradation limits long-term use; requires genipin crosslinking for enhanced mechanical stability; limited availability; and incomplete decellularization can lead to immune response	[37,38,39,40]
Chitosan	Polysaccharide from chitin	Promotes branching morphogenesis; interacts with endogenous growth factors; biocompatible; nontoxic; and biodegradable	Limited mechanical strength	[43]
Silk Fibroin Hydrogel	Silk fibroin from silk-producing arthropods	Customizable mechanical properties; photo-crosslinkable for controlled solidification in situ; and excellent biocompatibility	Long-term in vivo outcomes unknown and complex preparation process	[44]
Human Amniotic Membrane	Innermost layer of placenta (amnion)	Supports cell viability and differentiation and reduces inflammation and fibrosis	Lacks the ability to support vascularization and only grows on surfaces, hence not suitable for true 3D scaffolds	[51,52,53]
Collagen	Multiple, typically derived from animal sources	Promotes 3D structure formation and facilitates branching morphogenesis	Rapid degradation	[55,56,57,58]
Fibrin	Fibrinogen, usually from human plasma	Promotes cell adhesion, migration, vascularization, and nerve formation; supports epithelial tissue organization	Low mechanical strength and fast degradation rates, hindering long-term tissue support	[60]

### 3.2. Synthetic Biomaterials

In addition to natural biomaterials, synthetic biomaterials also considered for LG bioengineering [Table 2]. Long et al., (2006) first explored the feasibility of using polyethersulfone (PES) as a scaffold for LG bioengineering [62]. PES was fabricated using a phase inversion technique, carefully ensuring the material was semipermeable to allow for transfer of nutrients, such as ascorbic acid and glucose, while blocking rat IgG immunoglobulins to prevent immune reactions. Overall, PES tubes supported the attachment, growth, and proliferations of lacrimal acinar cells and allowed for selective permeability. However, the group explains that number of lacrimal acinar cell growth on PES was lower compared to other endothelial cells and had poor ability to proliferate in vitro when cultured over five times. However, the group suggests increasing the porosity of the material to increase cell growth. Moreover, the study conducted limited immunogenicity testing, and the overall immune response was not thoroughly examined because of the lack of in vivo testing.

Polyesters are also widely used in various medical and biological applications because of their biocompatibility, biodegradability, and good mechanical strength. Polyesters are mainly found in sutures, surgical meshes [63], orthopedic and ophthalmological implants [64], and drug delivery applications [65]. Selvam et al., (2007) developed a tissue-engineered tear secretory system by investigating cultured rabbit lacrimal acinar cell monolayers on polyester membrane scaffolds [66]. The functionality was assessed through active transepithelial ion fluxes across cell monolayers. The microporous polyester membrane supported proper cell polarity, tight junctions, functional protein secretion, and acinar cell proliferation. The authors highlighted the capability of polyester membrane scaffolds to form continuous monolayers, that exhibited transepithelial resistances characteristic of “leaky” epithelia observed in vivo. The group further suggests that these models will be useful in studying fundamental LG physiology. However, future research should prioritize in vivo studies to assess the biocompatibility and long-term stability of polyester.

Schrader et al., (2010) explored different scaffolds such as collagen, Matrigel, and polymeric material to determine the ideal substrate for cell adhesion, growth, and differentiation [67]. PLLA (poly-I-lactic acid) showed the best results for growth and morphological development of LG cells, and amniotic fluid proved to be a suitable carrier for LG cells for up to 21 days in vitro. While the study is a significant step toward demonstrating the potential of PLLA polymer for tissue repair, challenges such as maintenance of cell function, circumventing apoptosis, and ensuring the long-term viability of transplants remain. Selvam et al., (2007) also explored materials such as silicone, collagen I, copolymers of poly-D,L-lactide-co-glycolide (PLGA; 85:15 and 50:50), poly-L-lactic acid (PLLA), and Thermanox^®^ plastic cell culture coverslips for developing a bioengineered tear secretory system [68]. The group proposes that PLLA, with Matrigel coating, provided the best support for acinar cell-like morphology and contributed to the healthy formation of microvilli, secretory granules, and junctional complexes. PLLA is a synthetic biopolymer with modifiable mechanical properties derived from renewable resources such as cornstarch and sugar cane. PLLA is a slow, biodegradable scaffold material that turns into lactic acid, which may lead to local pH changes and inflammation. While PLLA’s biocompatible and mechanical properties are advantageous for creating a stable structure to support cell growth and tissue formation, PLLA’s hydrophobic nature may limit cell adhesion and protein absorption [69]. Selvam et al.’s (2007) group optimized PLLA’s properties through a Matrigel coating; however, they noted some loss in the mechanical integrity over an extended period of time. In contrast, silicon promoted the formation of 3D acinus-like structures and remained stable over an extended period of time. The group also demonstrated enhanced secretory functions with PLLA and, to a lesser extent, with PLGA, as indicated by β-hexosaminidase secretion, when compared to collagen I and plastic [68].

**Table 2 materials-17-05425-t002:** Synthetic biomaterials suitable for lacrimal gland tissue bioengineering.

Biomaterial	Features	Disadvantages	References
Polyethersulfone (PES)	Excellent mechanical stability; semipermeable and supports nutrient transfer while blocking immunoglobulins; and promotes acinar cell attachment	Hydrophobic surface; lower acinar cell proliferation compared to other endothelial cells; limited immunogenicity testing; and lack of vivo studies	[62]
Polyester	Supports cell polarity, tight junctions, protein secretion, and acinar cell proliferation	Lack of in vivo studies to assess the long-term biocompatibility and stability	[66]
Poly-I-lactic acid (PLLA)	Biodegradable; good mechanical properties; and supports acinar morphology, secretory granules, and junctional complexes	Very slow degradation, acidic byproducts may affect lacrimal gland cells; and a hydrophobic nature	[67,68]
Poly-D,L-lactide-co-glycolide (PLGA)	Biodegradable scaffold with good biocompatibility	Degrades into acidic byproducts and some loss in the mechanical integrity over time, with lower acinar secretory functions compared to PLLA	[67,68]
Silicon	Biocompatible and remains stable over an extended period of time	Non-biodegradable and requires further testing for long-term stability and potential immune responses in vivo	[68]

## 4. Cell Sources and Growth Factors

Stem cells serve as the repair system during injury and help replenish other cells, hence maintaining the structure and function of the LG. This regenerative property of LG, which is mediated by natural LG progenitor cells, are being investigated and integrated with bioengineered tissues to help facilitate advances in scaffold design and develop functional glandular tissues. Currently, limited research has been conducted on LG regeneration, and few cell sources have been explored. LG progenitor cells are found within the LG itself and are often isolated and expanded in vitro before being transplanted into an injured or diseased LG. Gromova et al., (2017) isolated and characterized epithelial cell progenitors from the LG of adult mice, specifically selecting a population with certain hallmark characteristics [70]. The isolated progenitor population expressed c-kit^+^dim/EpCAM^+^/Sca1^−^/CD34^−^/CD45^−^ and pluripotency factors Runx1 and EpCAM. During in vitro analysis, cells differentiated in structures resembling acini and ducts of the LG and were transplanted into both injured and diseased mouse models. Engraftment and integration into the LG were successful in the acute injury model, but, most notably, progenitor cell transplantation led to significant improvements in the LG structure of the chronic disease model, and tear production was significantly increased. Delcroix et al., (2023) further explored novel LG progenitor lines using the first transcriptomic atlas of LG in mice [71]. The group discovered Sox10+ cell populations, which are important for the creation of secretory units and contribute to the development of acinar, ductal, and myoepithelial lineages. Exploring novel progenitor cells expands the range of options for LG tissue engineering and enables the customization of therapeutic approaches to fit patient needs.

Regenerative tissue therapy has also been investigated through the use of mesenchymal stem cells (MSCs) in the repair and growth of injured LG [72]. MSCs are multipotent stromal cells that are able to differentiate into many cell types and are capable of self-renewal [73]. Jaffet et al., (2023) produced the first evidence of an MSC population within the human lacrimal gland [74]. These LG-MSCs possessed characteristics similar to MSCs from other tissues, notably expressing a higher level of IL-1β, which is involved in angiogenesis and LG development. This suggests that MSCs may be critical for maintaining the health of the LG and could play a role in its regeneration, opening new avenues for research at developing cell-based therapies. While these multipotential cells can support the growth of acinar and epithelial cells within the LG, it may be difficult to isolate a colony of these cells for clinical purposes [75]. Interestingly, MSCs derived from patients with Sjögren’s syndrome lacked certain colony-forming-unit efficiency and adipogenic differentiation potential compared to healthy controls [76]. This could offer valuable insight into the specific protocols for selecting MSCs for tissue regeneration.

Induced pluripotent stem cells (iPSCs) can also differentiate into multiple cell types, including acinar and ductal cells of the LG. Hayashi et al., (2022) developed a lacrimal-gland-like organoid using iPSCs and embryonic stem cells and demonstrated organoid maturation in rats as it developed lumina, expressed lactoferrin and lysozyme proteins, and contributed to tear-film production [77]. In addition, organoids showed an upregulation in genes associated with LG development such as BARX2 transcriptional factor. Asal et al., (2023) also utilized iPSCs to demonstrate its differentiation into acinar, ductal, and myoepithelial cell types, indicating its utility in LG bioengineering as a promising cell source [34]. This group employed a multizonal ocular differentiation strategy that efficiently produced LG-specific cells in just 4–7 weeks, achieving the shortest timeframe currently reported and paving the way for a more robust outlook.

Current studies explore new avenues of in situ LG regeneration, which include combining many novel strategies, such as adding growth factors and cytokines, gene therapy, and stem cell therapy. Lin et al., (2019) characterized human LG tissue through immunostaining progenitor markers [78]. Precursor cell markers C-Kit, K15, Nestin, and P63 were demonstrated to be involved in the differentiation of lacrimal epithelial cells into mini-glands. This exploration further advances stem-cell-based research on LG reconstruction and regeneration. However, the study reveals a rapid loss in differentiation in later cell passages, highlighting the need for research to maintain the differentiation capacity of these cells. Basova et al., (2017) investigated the role of Pannexin-1 in inflammation and the engraftment of epithelial progenitor cells in LG [79]. The authors demonstrated that controlling Panx1 activity can significantly improve engraftment and enhance the efficacy of regenerative treatments for LG dysfunction. Biomaterials that can effectively modulate the activity of such markers are crucial to producing localized anti-inflammatory effects and supporting transplant of LG tissue. Voronov et al., (2013) further investigated the role of specific Runt-related (Runx) transcription factors involved in the regulation, proliferation, and differentiation of stems cells in lacrimal bodies [80]. Specifically, the group revealed Runx1 and Runx2 are essential epithelial markers for LG morphogenesis, while Runx3 is expressed in both epithelial and mesenchymal compartments. As a result, the downregulation of such transcription factors significantly impairs LG growth, branching and development, highlighting the importance of research into specific factors required for LG tissue regeneration and biosynthesis in vivo. Recently, Finburgh et al., (2023) identified the role of fibroblast growth factor 10 (FGF10) in the development, homeostasis, and regeneration of LG [81]. In fact, the group found that the injection of FGF10 into damaged adult LG of mice significantly increased cell proliferation, cell viability, and accelerated the repair process in inflamed glands. FGF10 is found to be highly expressed in postnatal LG and to decrease in adulthood. Future research should conduct a large transcriptome analysis on the neonatal development of LG in mice and investigate growth factors largely active during this active stage in life. Growth factors implicated in LG regeneration should be added to scaffolds during the tissue-engineering process to allow for the optimal growth of damaged LG. Ueda et al., (2009) investigated the growth factors involved in postnatal LG development and demonstrated that EGF and HGF increased cell survival and proliferation, while the addition of FGF10 did not significantly stimulate epithelial cell proliferation in vitro [82]. However, FGF10 was more intensely expressed in mesenchymal cells than in epithelial cells. Figure 2 highlights the overall regeneration and reconstruction process of lacrimal gland restoration.

## 5. Advances in Tissue-Engineering Techniques

Tissue engineering utilizes diverse fabrication techniques, each with unique advantages and limitation. Although these techniques are widely discussed in other literature, lacrimal gland and soft tissue fabrication is frequently achieved using methods like electrospinning, 3D bioprinting, and electrospray techniques, and advancements in these areas are discussed.

Electrospinning applies a high voltage to a polymer solution, stretching the polymer into ultrafine fibers that are deposited onto a substrate to form a structured mesh (Figure 3) [83]. Certain polymers are more suited to electrospinning because of their ability to form stable jets and continuous fibers when subjected to an electric field. Synthetic materials such as polyethylene glycol (PEG) and poly-lactic-co-glycolic acid (PLGA) are largely utilized because of their relatively high molecular weights and dissolution in volatile solvents; however, a large variety of materials have been used because of the versatility of the electrospinning process [84]. Natural polymers, such as gelatin [85], cellulose [86], and silk [87], exhibit better biocompatibility; however, they are subject to faster degradation rates and fabrication difficulties. Synthetic materials are better able to withstand tension and shear forces and can be tailored to adapt to the required breakdown rate. However, they are limited by their low biocompatibility. Bosworth et al., (2021) incorporated decellularized ECM derived from porcine small intestinal submucosa and urinary bladder matrix with synthetic poly(ε-caprolactone) (PCL) to produce electrospun bioactive scaffolding that is used to support the stratification of large conjunctival defects [88]. The authors’ approach to creating the hybrid scaffolding combines the bioactivity of natural polymers with the structural integrity of synthetic polymers, allowing for the creation of fine-tuned fibrous structures. PCL nanofibers can be arranged randomly or with a straight, parallel orientation to exhibit unique tissue properties characteristic of natural ECM (Figure 4). Soscia et al., (2013) introduced a curvature to the electrospun PLGA nanofibers to support the growth and differentiation of salivary gland epithelial cells [89]. The authors propose that this setup enhanced the functional organization and polarity of epithelial cells and achieved a more realistic mimicry of the basement membrane environment in glandular structures. Other advancements in electrospinning, such as drug delivery methods utilizing multi-compartment nanofibers, have also been explored. Zhang et al., (2024) further advanced the electrospinning technique by generating the tri-chamber side-by-side electrospinning method, which allowed for the creation of distinct layers within the nanostructure and was capable of simultaneously delivering multiple drugs with distinct release profiles [90]. LG tissue engineering can leverage this nanostructural design to enhance the release of antioxidants and antibacterial agents, as well as to support tear duct system functions. Dong et al., (2024) developed a new type of core–shell nanofiber using coaxial electrospinning to enable the controlled release of ferulic acid, which has antioxidant and anti-inflammatory properties [91]. The structure allows for the precise and controlled release of the drug product. The group illustrates that this specific design effectively prevents the “negative tailing-off effect”, a phenomenon whereby the drug’s release from the carrier matrix gradually slows, resulting in sub-therapeutic drug levels and inconsistent dosing. Such scaffolds may help gradually deliver bioactive molecules essential for tissue repair and improve the effectiveness of LG-tissue-engineered implants.

Three-dimensional bioprinting has revolutionized tissue engineering by enabling the precise layering of biomaterials and cells to create complex scaffolds. Hydrogels, such as collagen, alginate, and decellularized porcine ECM solution, are commonly used as bioinks for gland tissue engineering. For example, Grumm et al., (2023) recently designed an alginate-porcine LG ECM scaffold that enables ion-based crosslinking through 3D bioprinting methods [94]. This further enabled the bioink to sustain a higher cell viability and density, while demonstrating larger sheer strength than control alginate bioink. Rodboon et al., (2022) introduced a novel protocol to engineer LG organoids that simulate aging and dysfunction using a magnetic 3D bioprinting platform (M3DB). The authors propose that the M3DB-based protocol is a consistent and scalable method for manufacturing LG organoids and demonstrates a reproducible approach to create functional models. This allows for further application in high-throughput platforms for testing drug formulations and gene therapies for DED [95]. Similarly, Ferreira et al., (2024) designed a study to create functional and aging models of LG organoids using M3DB platform [96]. Specifically, the team aimed to model DED by inducing cellular senescence and, subsequently, testing the efficacy of HMGB1-Box A gene therapy. Traditional models with Matrigel were reported to not be ideal for clinical translation because of its variability in models; however, the M3DB platform allowed for a user-friendly, scaffold-free, and homogeneous approach to high-throughput applications. Yin et al., (2023) demonstrated that microfluidics-based bioprinting provided superior spatial and temporal control when compared to traditional extrusion-based printing systems [97]. Traditional bioprinting methods face limitations in resolution and case shear stress damage, which is especially problematic for complex epithelial structures. In contrast, coaxial microfluidics allows for intricate structures using alginate-based microfibers and microtubes. Alginate enables rapid crosslinking, and the team highlights the usefulness of coaxial printheads to create branching structures and hollow tubes that mimic salivary gland features. As previously mentioned, crosslinking is a crucial technique for stabilizing hydrogels and other biomaterials in tissue engineering. In fact, genipin crosslinking has already shown to increase the mechanical stability and stiffness, as well as to optimize the degradability, in in vitro models of porcine decellularized LG [98].

Electrospraying is a popular technique in tissue engineering that uses an electric field to create a fine aerosol of small droplets containing cells, biomaterials, drug components, or other biomolecules that can be deposited on a scaffold to form tissues [83]. The precise droplet size and distribution may be controlled, allowing researchers to create complex tissue architectures with desired spatial arrangements and densities of compounds. Electrospraying is widely used in drug delivery systems because of its ability to sustain release over extended periods and enable targeted delivery, thereby enhancing therapeutic efficacy [99]. However, this technique requires the use of biomaterials in liquid form, which can restrict the range of biomaterials suitable for effective spraying. Additionally, certain cells and biomolecules may be sensitive to the preparation process, potentially affecting their overall activity and viability.

## 6. Future Directions

LG bioengineering is a nuanced and complex field that requires multiple considerations before developing viable therapies. The greatest challenge is mimicking native lacrimal gland architecture, as the LG is a highly specialized and intricate structure comprising multiple unique cells. The scaffold must support the appropriate spatial organization of different cell types to allow for correct tear production and flow. Furthermore, this involves the right stiffness, degradation profile, and porosity of the scaffold. A common difficulty among current studies is the ability to maintain cell function over time, and LG cell function seems to decrease as the distance from supporting cells or the scaffold increases. This loss of function may be due to the lack of a microenvironment similar to a natural LG away from the scaffold that would normally provide the necessary factors to sustain differentiated cells. There must also be further development in refining decellularization protocols to minimize residual antigens that may trigger immune reactions.

Furthermore, there seems to be no correct natural or synthetic biomaterial used for LG bioengineering. The native LG structure requires biomaterials with specific mechanical properties, which include optimal elasticity, mechanical strength, and degradation rate. Natural polymers offer good biocompatibility and cell viability profiles but lose structural integrity. Hence, more research must be conducted in developing hybrid biomaterials that involve both natural and synthetic polymers, as well as crosslinking and bioactive molecules to enhance cell adhesion and proliferation.

Another challenge lies within achieving proper vascularization and innervation of the implanted biomaterial. Multiple techniques can be employed to facilitate optimal blood supply throughout the LG. Lacrimal cells can be combined with endothelial cells that are derived from adult and human pluripotent stem cells [100]. Three-dimensional printing can be utilized as a scalable method for fabricating hydrogels with functional and hierarchical vascular architectures [101]. Furthermore, current tissue-engineering approaches have not addressed nervous system integration, which is crucial to control tear secretion. Further research must address methods on mimicking this neural control in vitro, possibly through the inclusion of neural cells or neurotrophic factors. Moreover, the engineered gland should respond appropriately to both sympathetic and parasympathetic neural signals when transplanted. This involves connecting the gland correctly to existing nerves and ensuring gland cells carry the required receptors and signaling pathways to respond to the release of neurotransmitters.

In addition, the manufacturing of scaffolds with precise micro- and nanostructures is technologically demanding. Three-dimensional printing techniques are a new emerging technology that offers significant advantages over conventional methods for many tissue-engineering applications. For example, stereolithography and selective laser sintering allows for the creation of highly complex and patient-specific scaffolds with precise control over scaffold parameters, such as porosity, size, and customizable cell placement [102]. Also, 3D printing techniques provide high repeatability and can be scaled to meet demand in the medical industry once the design and process are optimized. However, there are large regulatory hurdles in using scaffolds as biomedical devices by governing bodies. This is usually a costly and lengthy process in the commercialization of newly developed scaffolds. Rodboon at al., (2022) recently introduced a novel magnetic 3D bioprinting platform to rapidly and consistently biosynthesize lacrispheres from porcine primary LG cells for organoid creation [95]. The authors propose that the development of viable LG organoids through magnetic 3D printing has significant potential in LG transplantation, as well as drug discovery and screening, ultimately leading to new therapies for DED and other LG-related disorders. The magnetic 3D printing platform is described as xenogenic-free and highly scalable, using biocompatible magnetic nanoparticles to label cells and manipulate them using magnetic fields. This offers better control over the size and shape of the organoids, as well as the precise spatial organization of cells into 3D structures.

Lastly, tissue-engineered LGs must produce lacrimal fluid and successfully transport this fluid to the ocular surface. There is a lack of research addressing the connection of a tissue-engineered LG to a natural tear drainage system. New studies should investigate the creation of a synthetic duct that can be integrated in bioengineered LG or innovative micro-surgical procedures to enhance the ability to connect engineered ducts to existing ocular anatomy. For example, Holtmann et al., (2022) developed a proof-of-concept describing the feasibility of micro-anastomosis for transplantation of a human LG [103].

LG regeneration is also constrained by the challenges of proliferating cells in vitro. Advances in biomaterials that can provide a supportive environment and identify suitable long-term culture systems are crucial for successful cell transplantation. For example, Zeng et al., (2024) developed a serum-free culture strategy that utilizes a 2C small molecule combination consisting of both Y27632 and SB431542, which enhances and maintains proliferative capacity in vitro and provides a stable source of cells for further tissue engineering [36].

One of the key emerging trends in the field of LG bioengineering is the use of decellularized porcine ECM due to its high biocompatibility, ability to support cellular growth, and optimal features of natural ECM composition [98]. Although it offers some improvement over whole porcine lacrimal gland xenotransplantation, the decellularization process can still leave behind residual porcine antigens, potentially leading to an immune response. In addition, although rare, there is a risk of transmitting zoonotic pathogens from porcine tissues [104].

The new wave of tissue bioengineering is moving toward the addition of stem cells, such as MSCs and iPSCs, which can differentiate into acinar and ductal cells in the LG. Stem cells may be genetically modified to enhance their regenerative potential. For example, delivering proteins that encode for growth factors, anti-inflammatory proteins, or ECM components may improve the therapeutic outcome of LG regeneration. This possibility has been demonstrated, by Trousdale et al., (2005), through the expression of the TNF inhibitor gene in the LG, which led to the recovery of tear production and reduced inflammation in rabbits with induced autoimmune dacryoadenitis [105].

## 7. Conclusions

Advancements in biomaterials for lacrimal gland tissue engineering have opened promising pathways toward restoring natural tear production in patients with aqueous-deficient dry eye disease. Both natural and synthetic biomaterials have demonstrated significant potential in mimicking the extracellular matrix, supporting lacrimal gland cell proliferation and differentiation, and facilitating the formation of functional glandular structures. Natural biomaterials, such as Matrigel and decellularized lacrimal gland hydrogel, have shown particular efficacy in promoting acinar differentiation, while synthetic biomaterials like poly-I-lactic acid (PLLA) provide structural stability and support for long-term cell viability. The integration of stem cells and growth factors with these biomaterials further enhances regenerative outcomes, bringing us closer to viable therapeutic solutions.

While challenges remain in replicating the complex architecture of the lacrimal gland, achieving proper vascularization and innervation, and ensuring long-term functionality of engineered tissues, the rapid progress in biomaterial science and tissue-engineering techniques offers an optimistic outlook. Advances in electrospinning, microfluidics, and 3D bioprinting allow for the precise creation of scaffolds that can replicate the native tissue environment more closely, supporting cell organization and functional integration. Advances in 3D bioprinting and scaffold fabrication are particularly promising, providing innovative tools to overcome current limitations. With continued interdisciplinary research and collaboration, the development of effective biomaterial-based regenerative therapies for lacrimal gland dysfunction is within reach, holding the potential to significantly improve the quality of life for patients affected by this condition.

## Figures and Tables

**Figure 1 materials-17-05425-f001:**
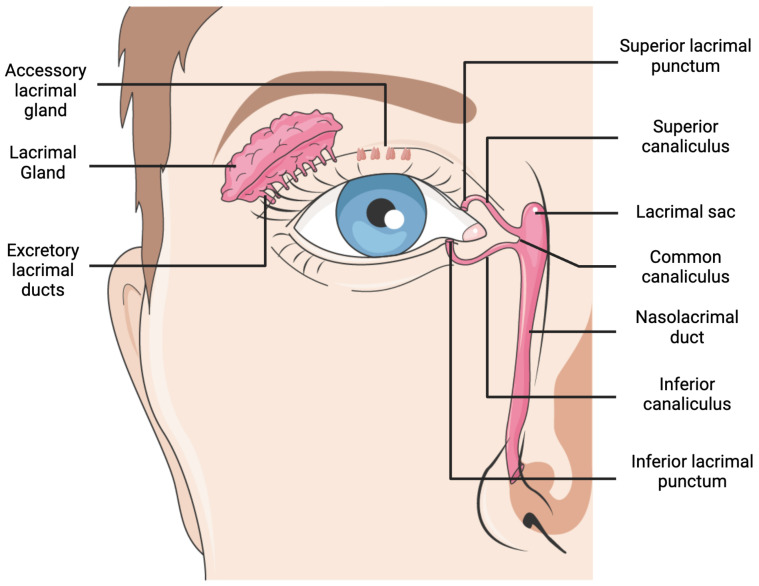
Lacrimal system anatomy. Created with BioRender.com and Servier Medical Art.

**Figure 2 materials-17-05425-f002:**
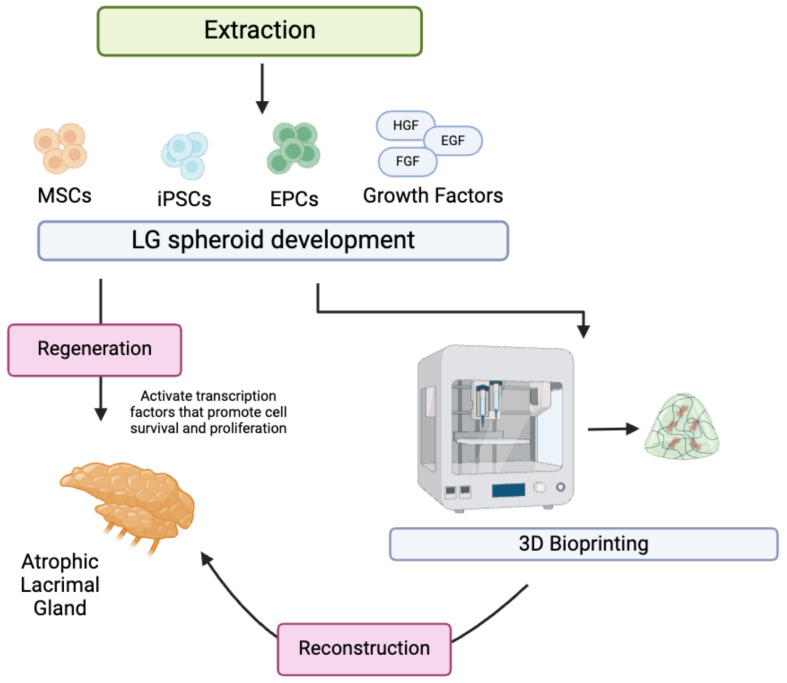
Regeneration and reconstruction of lacrimal gland involve the extraction of stem cells and subsequent spheroid development with added growth factors. Cultured spheroids may be integrated with scaffolds and combined with 3D bioprinting technology to reconstruct atrophic lacrimal gland. MSCs: mesenchymal stem cells; iPSCs: induced pluripotent stem cells; EPCs: endothelial progenitor cells; HGF: hepatocyte growth factor; EGF: epidermal growth factor; FGF: fibroblast growth factor. Created with BioRender.com.

**Figure 3 materials-17-05425-f003:**
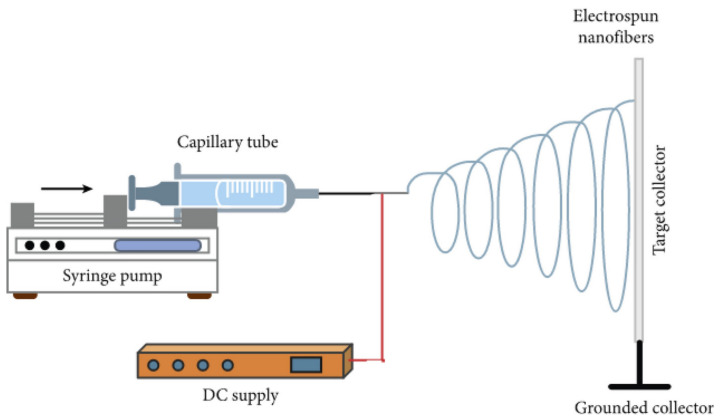
Electrospinning instrument setup. The bibliography consulted [92] is licensed under CC BY 2.0. To view a copy of this license, visit http://creativecommons.org/licenses/by/4.0/ (accessed on 28 October 2024).

**Figure 4 materials-17-05425-f004:**
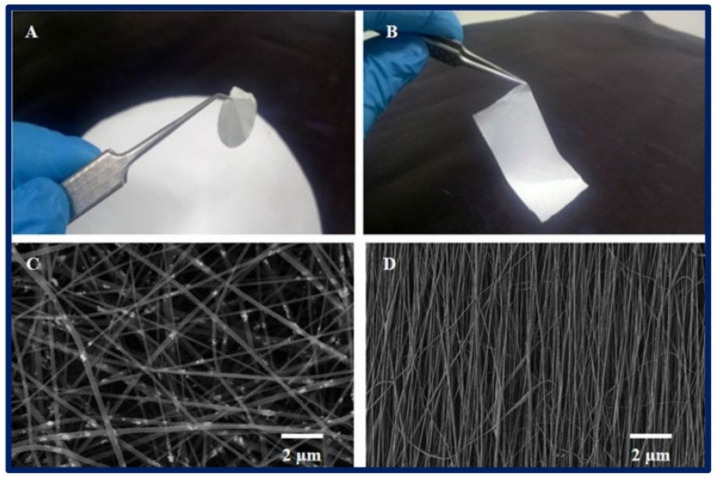
(**A**,**B**) Macroscopic view of a 2D poly-ε-caprolactone (PCL) scaffold fabricated by the electrospinning technique; (**C**,**D**) microscopic view of the scaffold under scanning electron microscopy; (**C**) nanofibers oriented randomly; (**D**) nanofibers with a straight, organized orientation. The bibliography consulted [93] is licensed under CC BY 2.0. To view a copy of this license, visit http://creativecommons.org/licenses/by/4.0/ (accessed on 28 October 2024).

## Data Availability

No new data were created or analyzed in this study.

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
