# Peer review of "Advanced Biomaterials for Lacrimal Tissue Engineering: A Review"

_materials, 2024, doi:10.3390/ma17225425_

Round 1

Reviewer 1 Report

Comments and Suggestions for Authors

Dear Authors,

You have faced an exciting topic related to human tissues as the lacrimal gland. As a review, the amount of pictures and references is OK. Your description goes from fundamental to specific cases which makes easier to understand

Many thanks for the effort

Author Response

Dear Reviewer,

Thank you very much for your thoughtful comments and positive feedback on our manuscript.

Regards, 

Reviewer 2 Report

Comments and Suggestions for Authors

The topic is interesting. However, the organizations of manuscript can be substantially improved. I recommend its acceptance for publication after some revision.

(1) The title may be “Advanced Biomaterials for Lacrimal Tissue Engineering: A review”.

(2) Section 3 and Section 4 are mainly about the raw biomaterials and active ingredients for treating Lacrimal disorders. It is recommended to increase a section about the materials conversion techniques that are explored to create advanced biomaterials from those reported in Section 3 and 4, such as 3DP, electrospinning (https://doi.org/10.1016/j.ijbiomac.2024.135753; https://doi.org/10.3390/polym16182614), and electrospraying (https://doi.org/10.3390/gels9090700) due to their capability of generating complex nanostructures.

(3) In the future directions section, information and issues that need to be resolved from a standpoint of Materials can be more.

(4) It should be “6. Conclusions”.

Author Response

Dear Reviewer,

Thank you for your thoughtful feedback and your positive assessment of our manuscript. We have carefully addressed each of your suggestions to improve the clarity and completeness of our paper. Below are our detailed responses to your comments:

  1. Title Suggestion
    We appreciate your suggestion to refine the title. We have updated it to “Advanced Biomaterials for Lacrimal Tissue Engineering: A Review,” as recommended, to better reflect the focus of the manuscript.
  2. Addition of a Section on Material Conversion Techniques
    Thank you for highlighting the importance of material conversion techniques, such as 3D printing, electrospinning, and electrospraying, in advancing biomaterials for lacrimal tissue engineering. We have included a dedicated section that briefly overviews these methods, referencing the sources you provided. This new section enhances the manuscript by showcasing the role of these techniques in generating complex nanostructures.
  3. Expansion on Future Directions
    Your suggestion to provide more detail on the unresolved issues from a “Materials” perspective in the Future Directions section is very valuable. We have expanded this section accordingly, incorporating a discussion on the key challenges and areas for future research to provide a more comprehensive outlook.
  4. Correction of Section Numbering
    Thank you for noting this correction. We have updated the numbering to reflect “6. Conclusions.”

We are grateful for your constructive feedback, which has helped us to strengthen the manuscript. We hope that our revisions meet your expectations and look forward to your assessment.

With warm regards,

Reviewer 3 Report

Comments and Suggestions for Authors

This review highlights recent advances in both natural and synthetic biomaterials used to support cell growth and scaffold the lacrimal gland (LG), demonstrating several strengths. Firstly, the logic is well-structured and coherent, making it easy for readers to follow the progression of ideas from the introduction to the conclusion. The review comprehensively covers a wide range of topics, including both natural and synthetic biomaterials, their properties, and their applications in lacrimal gland tissue engineering. Additionally, the inclusion of tables that compare the properties and performance of different biomaterials enhances the accessibility and utility of the information, making it easier for researchers and clinicians to reference and apply the findings in their work.

1.     Introduction: It would be beneficial to start with a brief overview of the prevalence and impact of aqueous-deficient DED to highlight the clinical significance of the research. This will help readers understand why regenerative therapies for the lacrimal gland (LG) are crucial. Also, more recent advances (VIEW 2023, 4, 20230026) may be included and discussed.

2.     Critical Analysis: Provide a critical analysis of the strengths and weaknesses of different biomaterials. For example, discuss the trade-offs between natural and synthetic materials in terms of biocompatibility, mechanical properties, and ease of fabrication.

3.     Mechanisms of Action: For each natural biomaterial discussed (e.g., Matrigel, decellularized extracellular matrices, chitosan, silk fibroin hydrogels, and human amniotic membrane), explain the mechanisms by which they support cell growth and differentiation. This will help readers understand the biological processes involved.

4.     3D Bioprinting: Highlight the latest advancements in 3D bioprinting and scaffold fabrication techniques. Discuss how these technologies can improve the precision and reproducibility of tissue engineering.

5.     Summary of Key Points: Summarize the key points discussed in the review, emphasizing the most significant advancements and their potential impact on the field.

Author Response

Dear Reviewer,

Thank you for your encouraging feedback and thoughtful recommendations. We appreciate your recognition of the strengths in our manuscript and have carefully addressed each of your suggestions to enhance its comprehensiveness. Please find our detailed responses below:

  1. Introduction
    We appreciate your suggestion to provide an overview of the prevalence and impact of aqueous-deficient dry eye disease (DED) to highlight the clinical significance of regenerative therapies for the lacrimal gland. We have now included a brief discussion on the prevalence of DED.
  2. Critical Analysis
    Thank you for recommending a critical analysis of the strengths and weaknesses of different biomaterials. We have incorporated discussions throughout the manuscript to compare natural and synthetic materials, addressing trade-offs such as biocompatibility, mechanical properties, and ease of fabrication.
  3. Mechanisms of Action
    We agree that including mechanisms of action for each natural biomaterial will enhance readers' understanding of the biological processes involved. We have now added a description of how each biomaterial (e.g., Matrigel, decellularized extracellular matrices, chitosan, silk fibroin hydrogels, and human amniotic membrane) supports cell growth and differentiation.
  4. 3D Bioprinting and Scaffold Fabrication Techniques
    Thank you for highlighting the importance of discussing 3D bioprinting and other scaffold fabrication advancements. We have introduced a separate section focused on the latest advancements in tissue engineering, with emphasis on 3D bioprinting, electrospinning, and electrospraying to enhance precision and reproducibility in scaffold fabrication.
  5. Summary of Key Points
    We have now added a concise summary section to emphasize the key advancements discussed and their potential impact on the field. This summary will provide readers with a clear understanding of the most significant insights from our review.

We appreciate your constructive feedback, which has helped us to further refine our manuscript. Thank you once again for your valuable input.

Warm regards,

Reviewer 4 Report

Comments and Suggestions for Authors

This is a review paper in a relatively narrow field, known as a 'niche'. Nevertheless, the authors have made every effort to introduce the research work in this field over the past five years, which shows that they are very familiar with the field. Thus, I would like to recommend this manuscript for publication after minor revision:

1. As a review article, the figures are so few. Please add more typical figures to the manuscript.

2. There are too many paragraphs in Section 3, some of which can be integrated together.

3. There are so many Key words. Please reduce the unnecessary words.

Comments on the Quality of English Language

The English could be improved to more clearly express the research.

Author Response

Dear Reviewer,

Thank you for your encouraging feedback and for recognizing our efforts in presenting recent research in this niche field. We are grateful for your insightful suggestions, and we have made revisions to address each point. Please see our responses below:

  1. Addition of Figures
    Thank you for your recommendation to add more figures. We have included two additional figures to enhance the visual representation of key concepts, making the information more accessible to readers.
  2. Integration of Paragraphs in Section 3
    We appreciate your comment regarding the organization of paragraphs in Section 3. We retained the current structure, as we feel it allows for clearer separation and a more detailed analysis of each biomaterial. This structure aims to provide in-depth insights into each biomaterial without overwhelming the reader.
  3. Reduction of Keywords
    Thank you for noting the number of keywords. We have reduced them by removing any redundant or unnecessary terms to improve the focus of the keyword list.

We sincerely appreciate your constructive feedback and hope that our revisions have enhanced the manuscript. Thank you again for your support and valuable input.

Warm regards,

Round 2

Reviewer 2 Report

Comments and Suggestions for Authors

The present edition can be accepted for publication!